# A Free Lunch in LLM Compression: Revisiting Retraining after Pruning

## Abstract

While Neural Network *pruning* typically requires retraining the model to recover pruning-induced performance degradation, state-of-the-art Large Language Model (LLM) pruning methods instead solve a layer-wise mask selection and reconstruction problem on a small set of calibration data to avoid full retraining, as it is considered computationally infeasible for LLMs. Reconstructing single matrices in isolation has favorable properties, such as convexity of the objective and significantly reduced memory requirements compared to full retraining. In practice, however, reconstruction is often implemented at coarser *granularities*, e.g., reconstructing a whole transformer block against its dense activations instead of a single matrix. In this work, we study the key design choices when reconstructing or retraining the remaining weights after pruning. We conduct an extensive computational study on state-of-the-art GPT architectures, and report several surprising findings that challenge common intuitions about retraining after pruning. In particular, we observe a *free lunch scenario*: reconstructing attention and MLP components separately within each transformer achieves the same model performance as more coarse-grained reconstructions while being the most resource-efficient. Most importantly, this Pareto-optimal setup achieves better performance than full retraining, despite requiring only a fraction of the memory. Furthermore, we demonstrate that simple and efficient pruning criteria such as Wanda can outperform much more complex approaches when the reconstruction step is properly executed, highlighting its importance. Our findings challenge the narrative that retraining should be avoided at all costs and provide important insights into post-pruning performance recovery for LLMs.

## 1 Introduction

LLMs have revolutionized Natural Language Processing (NLP) with state-of-the-art performance across a wide range of tasks from text generation to code synthesis. However, their scale comes with significant computational and memory demands, posing challenges for both researchers and practitioners. Model compression, particularly post-training pruning (Han et al., 2015; Gale et al., 2019; Hoefler et al., 2021), addresses these bottlenecks by identifying and removing redundant weights in pretrained Neural Networks (NNs), yielding *sparse* models with reduced inference costs.

While conventional pruning approaches typically require full retraining of the model to recover pruning-induced performance degradation, the drastic increase in model sizes has shifted the field's focus towards a more *local* perspective on the pruning problem, especially since full retraining of pruned LLMs is considered computationally prohibitive, if not infeasible (Sun et al., 2023; Frantar & Alistarh, 2023; Zimmer et al., 2023). In consequence, state-of-the-art approaches such as SparseGPT (Frantar & Alistarh, 2023) or Wanda (Sun et al., 2023) focus on solving a *layerwise pruning* problem and avoid retraining: Instead of finding a pruning mask and retraining the remaining weights in order to minimize a *global* loss, such approaches minimize a per-layer *local* loss, and consequently split the pruning problem into per-layer subproblems by pruning layers sequentially using a small set of calibration data. These methods are applied independently to each weight matrix in the model, and the final pruned model is then obtained by composing the individual pruned layers. Specifically, given a single layer with calibration input $X \in \mathbb{R}^{d_{in} \times B}$ and weights $W \in \mathbb{R}^{d_{out} \times d_{in}}$, the objective is

$$\min_{M, \hat{W}} \left\| WX - (M \odot \hat{W})X \right\|_F^2, \tag{1}$$

where $M \in \{0, 1\}^{d_{out} \times d_{in}}$ is a binary pruning mask satisfying some sparsity constraint, e.g., $\|M\|_0 \leq k$ for unstructured sparsity, $\hat{W} \in \mathbb{R}^{d_{out} \times d_{in}}$ is the matrix of reconstructed weights, and $\odot$ denotes the element-wise multiplication or Hadamard product. Here, $B$ denotes the total number of calibration tokens (number of sequences times sequence length). The problem of Equation 1 can be split into a *mask selection problem*, i.e., finding $M$ given fixed weights $W$, and a *reconstruction problem*, i.e., finding $\hat{W}$ given a fixed mask $M$. In this work, we focus on the latter. Reconstructing a single matrix in isolation is a convex quadratic problem, which admits an analytical solution, and requires significantly less memory compared to retraining the entire model. However, in practice, the reconstruction is often implemented at coarser *granularities*, e.g., reconstructing a submodel $f$ consisting of multiple layers (such as a transformer block) to fit its dense counterpart (e.g., Guo et al., 2024; Zimmer et al., 2023):

$$\min_{\hat{\theta}} \left\| f(X; \theta) - f(X; \hat{\theta}) \right\|_F^2, \tag{2}$$

where $f$ has parameters $\theta$ and $\hat{\theta}$ is restricted to the pruned parameterization.

In this work, we focus on the problem of reconstructing or retraining[1] the remaining weights of a pruned model, which we think is increasingly important as models become harder to prune and retraining-free pruning methods often may not suffice[2]. We systematically investigate key design choices for solving the reconstruction problem of Equation 1, including the choice of *propagation strategy* (how the calibration data defines inputs and targets), *loss function* (how reconstruction quality is measured), and the aforementioned granularity of reconstruction (the scope of the submodel that is reconstructed at once). Through an extensive computational study on state-of-the-art Generative Pretrained Transformer (GPT) architectures, we report several surprising findings that challenge common intuitions about retraining and reconstruction. Intuitively, increasing the reconstruction granularity should trade quality for resources: coarser granularity should improve final model quality but demand more resources for backpropagation, with full retraining being the most resource-intensive and per-matrix reconstruction the least. In fact, we find a sweet spot precisely when reconstructing the attention and MLP components of each transformer block separately. Surprisingly, this granularity is near the most resource-efficient yet achieves better perplexity and zero-shot accuracy than full-model retraining.

In other words, we first observe a *free-lunch scenario*: With the exception of per-matrix reconstruction, the reconstruction granularity does not have a significant impact on the final model performance, but finer granularities require far less memory and compute than full-model reconstruction or retraining, which makes them feasible even for massive LLMs. Second, our results reveal another surprising but consistent pattern: per-matrix reconstruction consistently underperforms, despite being the natural mathematical formulation of the reconstruction problem. Finally, we demonstrate that, when reconstruction is done properly, simple and efficient pruning methods like Wanda not only match but also outperform more complex approaches such as SparseGPT, even when both approaches undergo a reconstruction procedure after pruning.

**Contributions.** We summarize our main contributions as follows.

- **Analyzing LLM post-pruning reconstruction at scale.** We conduct an extensive computational study to systematically analyze reconstruction design choices, including propagation strategies (how calibration data defines inputs and targets), loss functions (how reconstruction quality is measured), and reconstruction granularities ranging from per-matrix to full-decoder.

- **Questioning the narrative of retraining infeasibility.** Our study highlights multiple surprising findings: First, per-matrix reconstruction consistently underperforms, despite corresponding to the most common formulation of the reconstruction problem. Second, there is a surprising sweet spot when it comes to the optimal setting for reconstruction: reconstructing attention and MLP components separately within each block is near the most

---

[1]We distinguish between *reconstruction* (local adaptation using intermediate activations as targets) and *retraining* (adaptation of the entire pruned model using true labels).

[2]While LLaMA-2-7B and LLaMA-3-8B achieve similar baseline perplexity on WikiText-2 (6.01 vs. 5.83), after pruning to 50% sparsity with Wanda, LLaMA-2 degrades to 6.71 while LLaMA-3 suffers a much larger drop to 8.96, hence making the pruned LLaMA-2 outperform its supposedly superior successor.

resource-efficient yet achieves the same performance as less granular reconstructions. Most importantly, this setup achieves much higher memory efficiency *and* better performance than full retraining, seemingly a free-lunch scenario for post-pruning reconstruction.

- **Highlighting the importance of (proper) reconstruction.** Leveraging these insights, once optimal local reconstruction is applied, we find that simple and efficient pruning criteria can outperform much more complex approaches, highlighting the importance of reconstruction, if done properly.

Together, these findings challenge the *avoid retraining at all costs* narrative. When done properly, local post-pruning reconstruction can deliver better perplexity and zero-shot accuracy than full-model retraining while using a fraction of the memory and compute, and it further enables simple pruning methods to match or surpass more complex mask-selection algorithms, highlighting the importance of reconstruction.

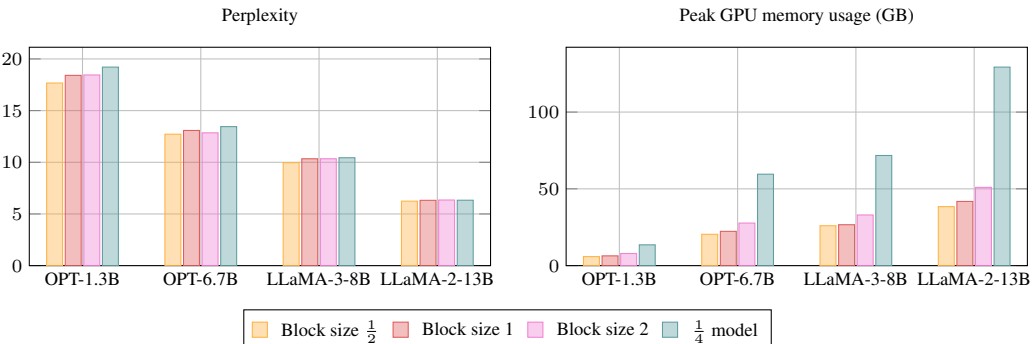

Figure 1: Perplexity and memory usage of four different models pruned to 2:4 sparsity with Wanda and reconstructed with different granularities. The number of calibration samples is 1024. $\frac{1}{4}$ of the model is 6, 8, 8, and 10 transformer blocks for OPT-1.3B, OPT-6.7B, LLaMA-3-8B, and LLaMA-2-13B, respectively. Displayed are the best results propagation strategies and loss functions.

## 2 BACKGROUND AND METHODOLOGY

We begin by surveying the existing literature on post-pruning reconstruction and related work. We then discuss the key design choices for local reconstruction, including the propagation strategy, loss function, and granularity.

### 2.1 BACKGROUND AND RELATED WORK

We discuss existing approaches to post-pruning reconstruction, with particular attention to the scope of the submodel that is reconstructed at once, which we term the *granularity*.

*Layer-wise pruning.* Both Wanda (Sun et al., 2023) and RIA (Zhang et al., 2024) are pure mask selection methods which operate at the per-matrix level. Wanda adapts magnitude pruning to LLMs by scaling the weight importance by the norm of their corresponding activations, and RIA extends this idea by further accounting for relative neuron importance. Frantar & Alistarh (2023), Zhao et al. (2024), and Boža (2024) integrate mask selection with weight reconstruction at the per-matrix level. SparseGPT (Frantar & Alistarh, 2023) builds upon the Optimal Brain Surgeon (OBS) (Hassibi et al., 1993) framework, which prunes one weight at a time and simultaneously adjusts the remaining weights based on the Hessian of the global loss using a greedy algorithm. SparseGPT makes this approach scalable for LLMs by solving the problem per matrix and using the local reconstruction objective instead of the global one, and pruning multiple weights simultaneously. FISTAPruner (Zhao et al., 2024) transforms the sparsity constraint into an $\ell_1$-norm penalty, solving the resulting problem via fast iterative shrinkage-thresholding. Boža (2024) introduces a method that alternates

between pruning weights using the Wanda criterion and reconstructing the remaining weights using the Alternating Direction Method of Multipliers (ADMM) method.

*Coarser-grained pruning.* BESA (Xu et al.) and LLM-BIP (Wu, 2024) perform mask selection for a transformer block using an optimization objective that allocates different amounts of sparsity to different submatrices within the block. EBFT (Guo et al., 2024) reconstructs entire transformer blocks and shows substantial improvements even when paired with strong pruning methods such as SparseGPT or Wanda. Shin et al. (2024) expand on this approach by introducing *global propagation*, which aims to better align the outputs of the pruned block with those of the original block. Instead of using local reconstruction, Zimmer et al. (2023); Muñoz et al. (2024) make full-model retraining feasible by using parameter-efficient fine-tuning methods like LoRA (Liu et al., 2020), which avoid the prohibitive costs associated with full-model retraining. SEFT (Xiao et al., 2025) builds on this strategy by dynamically evolving pruning masks during fine-tuning, allowing the sparsity patterns to adapt over time, akin to dynamic sparse training (Liu et al., 2020). Lastly, the importance of granularity has already been highlighted in the context of quantization. Li et al. (2021) initially demonstrated that block-wise reconstruction following quantization outperforms both layer-wise and full-model strategies for Convolutional Neural Networks (CNNs). Jeon et al. (2022) extended this approach using non-uniform quantization and applying it to transformer architectures.

## 2.2 KEY DESIGN CHOICES WHEN RECONSTRUCTING AFTER PRUNING

We analyze three crucial design choices for local reconstruction: the *propagation strategy*, which determines how calibration data is transmitted through the model; the *loss function*, by which reconstruction quality is measured; and the *granularity*, which specifies the size of the submodel reconstructed at once.

**Propagation: How to define inputs and targets?** Pruning a model locally involves splitting it into submodels and sequentially pruning each submodel. Before discussing the different propagation strategies, recall the pruning objective in Equation 2 and let us specify how the input activations $\mathbf{X}$ for the submodel $f$ we want to prune are computed. We denote by $\mathbf{X} := f_0(\mathbf{X}_0; \theta_0)$ the output of the submodel $f_0$ consisting of all layers prior to $f$ with parameters $\theta_0$ based on input data $\mathbf{X}_0$. Further, we denote by $\hat{\mathbf{X}} := f_0(\mathbf{X}_0; \hat{\theta}_0)$ the output of the submodel $f_0$ using the pruned weights $\hat{\theta}_0$. The different propagation strategies are defined by whether we use calibration inputs $\hat{\mathbf{X}}$ or $\mathbf{X}$, and similarly, whether to align the outputs with $f(\mathbf{X}; \theta)$ or $f(\hat{\mathbf{X}}; \theta)$. We outline three distinct strategies:

1. Dense Propagation (DP): Both inputs and targets are sourced from the original dense model, i.e., the objective is $\min_{\hat{\theta}} \left\| f(\mathbf{X}; \hat{\theta}) - f(\mathbf{X}; \theta) \right\|_F^2$. Here, the pruned submodel is reconstructed to best match the dense one, ignoring the error caused by pruning the prior submodels.

2. Sparse Propagation (SP): Inputs come from the pruned model and targets are generated by further processing the sparse inputs through the dense model, i.e., the objective is $\min_{\hat{\theta}} \left\| f(\hat{\mathbf{X}}; \hat{\theta}) - f(\hat{\mathbf{X}}; \theta) \right\|_F^2$. This approach is consistent with the EBFT method (Guo et al., 2024), where the reconstruction objective takes into account the error caused by pruning the prior submodels.

3. Mixed Propagation (MP): Inputs come from the pruned model, while targets are sourced from the dense model, i.e., the objective is $\min_{\hat{\theta}} \left\| f(\hat{\mathbf{X}}; \hat{\theta}) - f(\mathbf{X}; \theta) \right\|_F^2$. This approach tries to correct for the error caused by pruning the prior submodels by steering the activations to those of the dense model. Shin et al. (2024) refer to this strategy as *global propagation*.

**Loss: How to measure reconstruction quality?** The loss function measures the similarity between the outputs of the reconstructed submodel and the target activations. We consider two loss functions: Mean Squared Error (MSE), the squared Euclidean distance averaged over batches; and Cosine Similarity (CS), a well-known measure of similarity between latent representations in NLP (Mikolov et al., 2013; Gromov et al., 2024).

**Granularity: How to split the model into submodels?** The reconstruction granularity determines how many layers are reconstructed jointly. In the following, a transformer block refers to the attention

component followed by the feed-forward component. We explore five different reconstruction granularities:

i) *Per-matrix*: Each weight matrix is reconstructed independently, cf. Equation 1.

ii) *Block size one-half*: Attention and feed-forward components of one block are reconstructed separately.

iii) *Block size one*: Each transformer block is reconstructed independently, cf. Guo et al. (2024).

iv) *Multi-block*: Block sizes greater than one but smaller than the total number of transformer blocks in the model.

v) *Full decoder*: All transformer blocks are reconstructed simultaneously.

The full decoder approach resembles full-model distillation, except that the embedding layer remains frozen, and the Language Modeling (LM) head is excluded from the loss calculation, as the reconstruction objective is to align the output of the final transformer block with the target activations.

## 3 EXPERIMENTS

We perform experiments on OPT-1.3B, OPT-6.7B, LLaMA-2-13B, LLaMA-3-8B, and Qwen-2.5-32B-Instruct (Zhang et al., 2022; Touvron et al., 2023; Grattafiori et al., 2024; Yang et al., 2024), using sequences sampled from C4 (Raffel et al., 2020) as calibration data. Models are evaluated on perplexity (lower is better) on WikiText-2 (Merity et al., 2016) and average zero-shot accuracy (higher is better) using the EleutherAI evaluation suite (Gao et al., 2024) (accuracies for each task can be found in Appendix B). Unless stated otherwise, we use one-shot layer-wise pruning with Wanda (Sun et al., 2023), considering both unstructured sparsity and $N{:}M$ semi-structured patterns (Mishra et al., 2021). Following Sun et al. (2023), all linear layers except the embedding and final LM head are pruned with uniform sparsity. For each local reconstruction configuration, we sweep learning rates in $[10^{-6}, 10^{-3}]$ with a linear schedule and 10% warm-up, and the number of epochs in $\{1, 5, 10, 20\}$. Unless stated otherwise, we use 1024 calibration samples for both pruning and reconstruction. For LoRA fine-tuning after pruning, we use Mask-LoRA (Zimmer et al., 2023). Retraining, fine-tuning, and reconstruction use AdamW (Loshchilov & Hutter, 2019) with a batch size of two. A detailed hyperparameter table can be found in Appendix C. The results are averaged over multiple random seeds, and our code will be publicly released to ensure reproducibility.

### 3.1 PROPAGATION STRATEGIES AND LOSS FUNCTIONS HAVE NO SYSTEMATIC IMPACT.

As an additional metric, we define *recovery* as the fraction of the pruned-to-dense perplexity gap closed by reconstruction:

$$\text{recovery} = \frac{\text{PPL}_{\text{pruned}} - \text{PPL}_{\text{reconstructed}}}{\text{PPL}_{\text{pruned}} - \text{PPL}_{\text{dense}}}. \tag{3}$$

Recovery measures the improvement achieved by reconstruction, normalized by the difference between the pruned and dense perplexity. Recovery equals one when reconstruction fully restores dense performance, zero when no improvement occurs, and negative when reconstruction degrades performance.

Before presenting our main results, we justify our default hyperparameter choices through ablation studies. Perplexity decreases up to approximately 20 epochs for all local reconstruction approaches (see Figure 3). A learning rate sweep over $[10^{-6}, 10^{-3}]$ reveals that each model has an optimal learning rate in this range (see Figure 4 in the appendix).

Figure 2 summarizes the impact of propagation strategies (MP, SP, DP) and loss functions (MSE, CS) on recovery (defined in Equation 3). Each panel shows histograms comparing two methods, indicating how often one outperforms the other and by how much. The horizontal axis represents recovery between the two options (values greater than zero mean the option named first did better; values less than zero mean it did worse). The vertical axis counts runs in each difference bin. The orange bars compare the two options while keeping everything else the same (same model, sparsity, reconstruction granularity, learning rate/epochs search, and propagation strategy or loss function

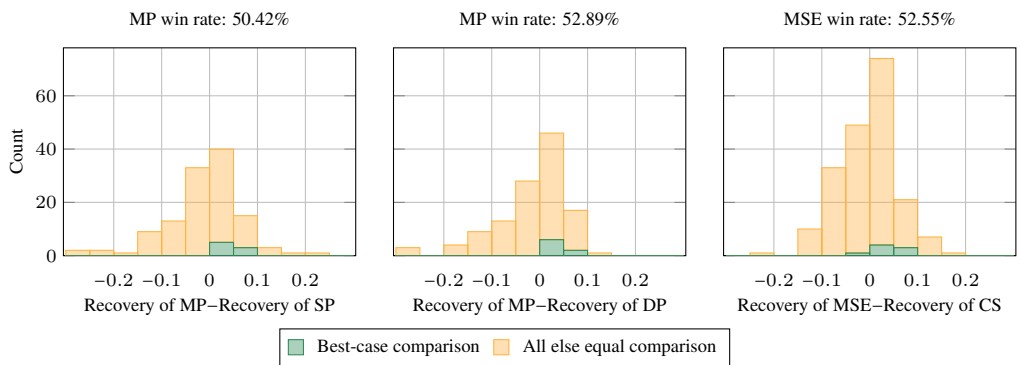

Figure 2: Histograms of recovery differences (Equation 3) between methods. Orange bars compare methods under identical hyperparameter configurations. Green bars compare each method's optimal setting for a given model and sparsity. The data used to generate these plots is shown in Table 6, Table 8, Table 9, and Table 10 in Appendix B.

Table 1: Comparison of local reconstruction (block size 1) and Mask-LoRA fine-tuning on LLaMA-3-8B when pruned with Wanda to 50% unstructured sparsity. The compute time is for a single NVIDIA H200 GPU.

| Method | Number of samples | Perplexity | Compute Time |
|---|---|---|---|
| Local reconstruction | 1024 | 7.72 | 42.36 minutes |
| Mask-LoRA fine-tuning | 131072 | 7.70 | 19.17 hours |

depending on what is being compared). Green bars compare each method's optimal configuration for a given model and sparsity.

The orange bars show that no propagation strategy or loss function consistently outperforms others when hyperparameters are identical. The green bars indicate that MP tends to achieve the highest recovery across propagation strategies, and MSE is more often favorable than CS. However, these effects are small and have little practical impact on final model quality. For the remainder of our experiments, we therefore focus on reconstruction with MP and MSE loss. Full results for all propagation strategies and loss functions are reported in Appendix B.

Figure 4 shows that the optimal learning rate is stable across models and sparsity types. For reconstruction with block size $\frac{1}{2}$ under MP and MSE loss, we consistently observe the optimal learning rate near $3 \times 10^{-5}$ across all four models (OPT-1.3B/6.7B, LLaMA-2-13B, LLaMA-3-8B) and both unstructured and semi-structured sparsity. This stability reduces the need for extensive grid searches in practice.

### 3.2 GRANULARITY MATTERS: CHALLENGING THE AVOID-RETRAINING-AT-ALL-COSTS NARRATIVE

We next analyze the effect of reconstruction granularity and present several surprising findings.

**Per-matrix reconstruction consistently underperforms, while local reconstruction outperforms full retraining.** Table 3 shows full results for OPT-6.7B reconstructed at different granularities. As stated, we now mainly focus on the columns corresponding to MP propagation and MSE loss. Layer norm parameters are fixed across all settings since they are not updated by per-matrix reconstruction. For 2:4 sparsity, the best per-matrix reconstructed model performs worse in perplexity and zero-shot accuracy than the worst model using any other granularity. For both sparsity types, MP reconstruction collapses completely for per-matrix approaches, while coarser granularities perform well. This

Table 2: Perplexity and zero-shot accuracy of four different models pruned to 50% unstructured and 2:4 sparsity with Wanda and reconstructed with MSE and MP. The best result for each (sparsity type, model) combination is highlighted in bold. ↓: lower is better, ↑: higher is better.

| Model Dense PPL | Block size | Perplexity ↓ | | Zero-shot accuracy (in %) ↑ | |
|---|---|---|---|---|---|
| | | unstructured | 2:4 | unstructured | 2:4 |
| LLaMA-3-8B 5.83 | No rec. | 8.96 | 21.85 | 56.89 | 45.67 |
| | $\frac{1}{2}$ | 7.79 | **10.24** | **59.77** | **55.23** |
| | 1 | **7.72** | 10.31 | 58.67 | 54.44 |
| | 2 | 7.75 | 10.32 | 58.36 | 52.72 |
| | 8 | 7.89 | 10.53 | 57.90 | 52.87 |
| LLaMA-2-13B 4.57 | No rec. | 5.54 | 8.39 | 60.25 | 53.23 |
| | $\frac{1}{2}$ | **5.25** | **6.22** | 61.14 | **58.77** |
| | 1 | **5.25** | 6.29 | **61.43** | 58.65 |
| | 2 | 5.27 | 6.35 | 60.86 | 57.84 |
| | 10 | 5.28 | 6.32 | 61.06 | 57.11 |
| Qwen-2.5-32B 4.90 | No rec. | 6.09 | 8.00 | 68.03 | 66.41 |
| | $\frac{1}{2}$ | 5.93 | 6.06 | 69.05 | 68.79 |
| | 1 | **5.92** | **5.97** | **69.18** | 69.06 |
| | 2 | 5.98 | 6.01 | 69.15 | **69.19** |
| | 8 | 5.97 | 7.07 | 69.09 | 68.86 |

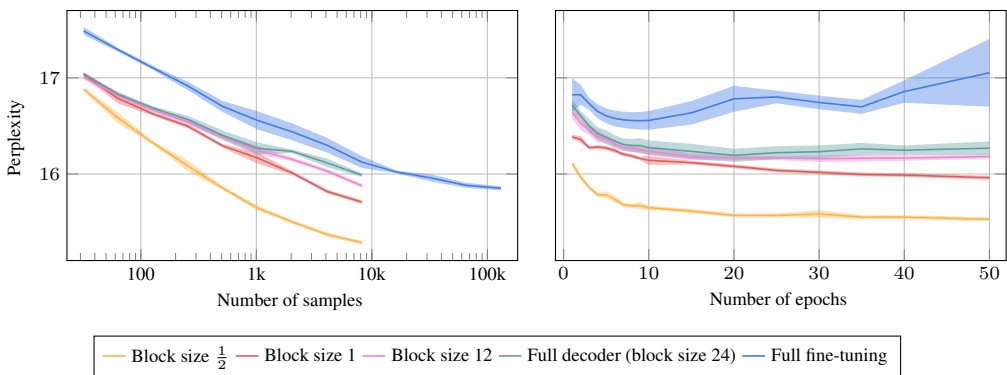

Figure 3: OPT-1.3B pruned to 50% unstructured sparsity with Wanda, showing mean perplexity vs. number of calibration samples (left) and number of epochs (right). The shaded areas indicate the min-max range over random seeds. Reconstruction uses MSE loss and DP. Full fine-tuning uses cross-entropy loss and true labels.

suggests that individual matrices lack the capacity to absorb pruning-induced errors from earlier layers. Similar trends hold across different models (see Appendix B).

Most surprisingly, full retraining consistently yields the worst perplexity across calibration sizes and epochs in Figure 3, underperforming local reconstruction approaches. Even when increasing the calibration set size to 131k samples, full retraining OPT-1.3B yields higher perplexity than reconstruction with block size $\frac{1}{2}$ and 1024 samples (15.85 vs. 15.60 perplexity). On LLaMA-3-8B, Mask-LoRA (Zimmer et al., 2023) fine-tuning achieves comparable perplexity to local reconstruction when using 128 times more samples and 27.2 times more compute time (see Table 1).

Table 3: Perplexity and zero-shot accuracy of OPT-6.7B pruned to different sparsity types with Wanda and reconstructed at various granularities. Layer norm parameters are fixed for comparability with per-matrix reconstruction, the layer norm parameters are fixed in every setting. The best result for each setting is underlined, the best result for each sparsity type is highlighted in bold. ↓: lower is better, ↑: higher is better. Dense baseline: 10.86 perplexity, pruned unstructured: 12.06, pruned 2:4: 16.05.

| Sparsity type | Block size | Perplexity ↓ | | | | | | Zero-shot accuracy (in %) ↑ | | | | | |
| | | SP | | MP | | DP | | SP | | MP | | DP | |
| | | MSE | CS | MSE | CS | MSE | CS | MSE | CS | MSE | CS | MSE | CS |
| unstructured | Per-matrix | 11.55 | 11.72 | 98.15 | 49.72 | 11.96 | 11.99 | 49.39 | 49.48 | 43.64 | 44.95 | 48.78 | 48.98 |
| | $\frac{1}{2}$ | 11.49 | **11.27** | 11.66 | 11.34 | 11.51 | 11.40 | 49.99 | **50.62** | 50.60 | 50.49 | 49.86 | 50.14 |
| | 1 | 11.65 | 11.59 | 11.95 | 11.90 | 11.56 | 11.71 | 49.85 | 50.54 | 49.97 | 50.27 | 49.86 | 50.61 |
| | 2 | 11.61 | 11.56 | 11.78 | 11.84 | 11.60 | 11.57 | 50.02 | 50.52 | 50.23 | 50.18 | 49.92 | **50.62** |
| | 8 | 11.84 | 11.90 | 11.86 | 11.91 | 11.88 | 11.93 | 49.73 | 49.75 | 50.02 | 49.89 | 49.84 | 49.68 |
| 2:4 | Per-matrix | 14.25 | 15.82 | 293.26 | 104.03 | 16.06 | 15.52 | 47.02 | 46.94 | 38.42 | 40.62 | 45.53 | 45.99 |
| | $\frac{1}{2}$ | 13.39 | 12.98 | 12.97 | **12.72** | 13.45 | 12.97 | 47.87 | 48.18 | **48.90** | 48.80 | 47.84 | 48.28 |
| | 1 | 13.29 | 13.04 | 13.85 | 13.58 | 13.59 | 13.21 | 47.91 | 48.61 | 48.31 | 48.59 | 47.84 | 48.76 |
| | 2 | 13.43 | 12.93 | 13.46 | 13.40 | 13.41 | 12.90 | 47.83 | 48.47 | 48.48 | 48.78 | 48.03 | 48.75 |
| | 8 | 13.47 | 13.51 | 13.52 | 13.55 | 13.64 | 13.24 | 47.92 | 47.98 | 48.35 | 48.31 | 47.82 | 48.25 |

Table 4: Perplexity and zero-shot accuracy of LLaMA-2-7B when pruned with structured Wanda (Wanda-sp) and FLAP. The reconstruction after pruning with Wanda-sp is performed with MP and MSE at block size 1.

| Sparsity | Perplexity | | | Zero-shot accuracy | | |
| | FLAP | Wanda-sp no rec. | Wanda-sp rec. | FLAP | Wanda-sp no rec. | Wanda-sp rec |
| 20% | 7.81 | 8.61 | 7.76 | 53.42% | 54.40% | 56.09% |
| 50% | 46.25 | 57.77 | 30.97 | 42.87% | 36.05% | 44.24% |

**Reconstructing at block size $\frac{1}{2}$ is most effective.** For OPT-1.3B in Figure 3, block size $\frac{1}{2}$ (reconstructing attention and MLP separately) consistently achieves the lowest perplexity, while block sizes 1, 12, and 24 yield essentially identical performance. For OPT-6.7B (Table 12 in the appendix), block size $\frac{1}{2}$ remains superior to coarser granularities, though the gap is smaller. For LLaMA-2-13B, LLaMA-3-8B, and Qwen-2.5-32B (Table 2), all block sizes perform nearly identically in both perplexity and zero-shot accuracy, with no clear advantage for any particular granularity. For more detailed results and experiments on the MiniPile data set (Kaddour, 2023), see Appendix B. Meanwhile, Figure 1 shows that peak memory usage grows substantially with block size, meaning finer granularities achieve comparable quality at lower memory cost. Together, these results indicate that block size $\frac{1}{2}$ provides the best trade-off: equal or better performance at lower computational cost.

### 3.3 RECONSTRUCTION ENABLES SIMPLE PRUNING METHODS TO OUTPERFORM COMPLEX ONES

Table 4 compares structured Wanda (Wanda-sp) before and after reconstruction with FLAP (An et al., 2024). Compared to the simple uniform structured pruning performed by Wanda-sp, FLAP is a complex pruning method that non-uniformly distributes sparsity across transformer blocks to achieve superior performance. Table 5 compares magnitude pruning, Wanda, and SparseGPT on LLaMA-2-13B, LLaMA-3-8B, and Qwen-2.5-32B-Instruct under block size $\frac{1}{2}$ with MP and MSE loss. Magnitude pruning and Wanda are relatively simple approaches: magnitude pruning requires no calibration data and prunes based solely on weight magnitudes, while Wanda extends this with a single forward pass to scale importance by activation norms. In contrast, SparseGPT is more complex, alternating between mask selection and Hessian-based weight updates at the per-matrix level.

Two consistent patterns emerge. First, pruning difficulty changes across model families: magnitude pruning without reconstruction yields perplexity comparable to Wanda and SparseGPT for LLaMA-2-13B, but performs worse on Qwen-2.5-32B and collapses entirely for LLaMA-3-8B. Second, local reconstruction substantially improves simple methods. On LLaMA-2-7B, structured

Table 5: Perplexity of LLaMA-2-13B, LLaMA-3-8B, and Qwen-2.5-32B when pruned with different pruning methods under four sparsity types and reconstructed with MP and MSE at block size $\frac{1}{2}$. The best result for each model and sparsity type is highlighted in bold.

| Model | Sparsity type | SparseGPT | | Wanda | | Magnitude | |
|---|---|---|---|---|---|---|---|
| | | No rec. | Rec. | No rec. | Rec. | No rec. | Rec. |
| LLaMA-2-13B | 50% unstructured | 5.52 | 5.29 | 5.54 | **5.25** | 6.37 | 5.36 |
| | 60% unstructured | 7.46 | 6.83 | 7.78 | **6.25** | 8.21 | 6.42 |
| | 2:4 | 8.04 | 6.37 | 8.38 | **6.22** | 8.45 | 6.50 |
| | 4:8 | 6.45 | 5.79 | 6.54 | **5.72** | 6.75 | 5.93 |
| LLaMA-3-8B | 50% unstructured | 8.67 | 7.96 | 9.01 | **7.78** | 152.11 | 13.18 |
| | 60% unstructured | 14.12 | 9.43 | 19.38 | **8.91** | 823.65 | 17.46 |
| | 2:4 | 14.46 | 10.09 | 21.82 | **9.94** | 765.06 | 26.22 |
| | 4:8 | 11.08 | 9.84 | 12.76 | **9.42** | 337.78 | 18.16 |
| Qwen-2.5-32B | 50% unstructured | 6.21 | 6.15 | 6.09 | **6.03** | 27.80 | 6.26 |
| | 60% unstructured | 7.48 | **7.22** | 7.62 | 7.24 | 90.94 | 7.84 |
| | 2:4 | 7.81 | 7.44 | 8.00 | **7.36** | 36.60 | 7.59 |
| | 4:8 | 6.93 | 6.77 | 6.80 | **6.64** | 34.62 | 6.82 |

Wanda with reconstruction outperforms FLAP, even though, especially for 50% sparsity, structured Wanda significantly underpreforms without reconstruction. On LLaMA-2-13B and Qwen-2.5-32B, magnitude pruning with reconstruction not only recovers performance but matches or even surpasses SparseGPT under both unstructured and semi-structured sparsity. Wanda with reconstruction achieves further gains, consistently outperforming SparseGPT on all three models. Since SparseGPT combines mask selection with second-order updates, these results show that its algorithmic complexity provides no advantage once local reconstruction is applied—Wanda then achieves superior performance.

## 4 CONCLUSION

In this work, we revisited post-pruning retraining of LLMs through the lens of local reconstruction. Across models, we consistently observe that reconstructing attention and MLP components separately within each transformer block (block size $\frac{1}{2}$) is Pareto-optimal: it matches or outperforms coarser granularities in perplexity and zero-shot accuracy while requiring only a fraction of the memory. We further ablate propagation strategies and loss functions. When hyperparameters are identical, no strategy (SP, MP, DP) or loss function (MSE, CS) universally outperforms others; in optimal-configuration comparisons, MP paired with MSE tends to have a slight advantage. Finally, once proper local reconstruction is applied, simple mask-selection methods such as Wanda can match or surpass more complex algorithms like SparseGPT across sparsity regimes, underscoring the central role of the reconstruction step.

Taken together, our results suggest a practical recipe for post-pruning recovery in LLMs: favor block size $\frac{1}{2}$ local reconstruction, pair it with MP and MSE, and use simple pruning methods when possible. This configuration is simple, memory-efficient, and reliably competitive, outperforming full-model fine-tuning on the same calibration data in our evaluations.

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

## A  USE OF LARGE LANGUAGE MODELS

Large language models were used to aid in writing (polishing text), retrieving related work, generating code for plots, and implementing standard components. No novel research ideas or results were produced by LLMs.

## B  ADDITIONAL EXPERIMENTS

In this section, we present additional experiments with OPT-1.3B, OPT-6.7B, LLaMA-2-13B, and LLaMA-3-8B models.

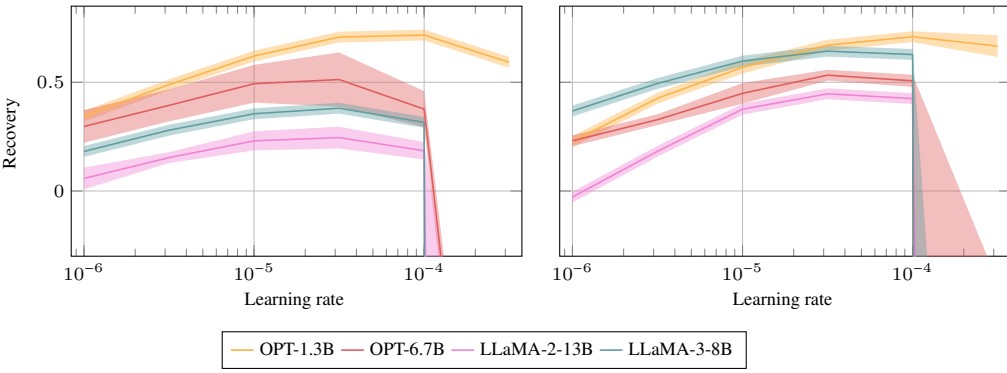

Figure 4: Learning rate ablation for reconstruction after pruning with Wanda to 50% unstructured sparsity (left) and 2:4 semi-structured sparsity (right). The $y$-axis is the recovery defined in Equation 3. Reconstruction was performed with block size $\frac{1}{2}$, MP, and MSE loss.

Table 6: Perplexity and zero-shot accuracy of OPT-1.3B pruned to different sparsity types with Wanda and reconstructed in different settings with 1024 calibration samples on a single random seed. The best result for each setting is underlined, the best result for each sparsity type is highlighted in bold. SP, MP, DP: sparse, mixed, and dense propagation. CS: cosine similarity. ↓: lower is better, ↑: higher is better. Base perplexity of OPT-1.3B dense: 14.62, pruned unstructured: 18.31, pruned 2:4: 28.11.

| Sparsity type | Block size | Perplexity ↓ | | | | | | Zero-shot accuracy (in %) ↑ | | | | | |
| | | SP | | MP | | DP | | SP | | MP | | DP | |
| | | MSE | CS | MSE | CS | MSE | CS | MSE | CS | MSE | CS | MSE | CS |
|---|---|---|---|---|---|---|---|---|---|---|---|---|---|
| 50% unstructured | $\frac{1}{2}$ | 15.60 | 15.45 | 15.47 | **15.43** | 15.60 | 15.45 | 44.99 | **45.16** | 44.63 | 44.99 | 44.68 | 45.02 |
| | 1 | 16.01 | 15.86 | 16.42 | 16.52 | 16.07 | 16.12 | 44.03 | 44.63 | 44.08 | 44.32 | 44.11 | 44.78 |
| | 2 | 16.03 | 15.81 | 16.26 | 16.34 | 16.03 | 15.78 | 43.86 | 44.47 | 44.26 | 44.27 | 44.00 | 44.49 |
| | 6 | 16.12 | 16.17 | 16.17 | 16.25 | 16.14 | 16.20 | 44.38 | 44.35 | 44.27 | 44.41 | 44.51 | 44.29 |
| | 12 | 16.20 | 16.34 | 16.19 | 16.28 | 16.22 | 16.38 | 43.99 | 44.02 | 44.11 | 44.33 | 44.26 | 44.03 |
| | 24 | 16.29 | 16.44 | 16.29 | 16.42 | 16.31 | 16.43 | 44.24 | 44.34 | 44.22 | 44.38 | 44.32 | 44.27 |
| 2:4 | $\frac{1}{2}$ | 18.49 | 17.96 | **17.67** | 17.82 | 18.52 | 18.04 | 43.39 | 43.39 | 44.09 | **44.17** | 43.50 | 43.56 |
| | 1 | 19.39 | 18.41 | 19.37 | 19.74 | 19.67 | 18.94 | 42.67 | 43.53 | 42.97 | 43.27 | 42.66 | 43.64 |
| | 2 | 19.33 | 18.50 | 18.96 | 19.25 | 19.37 | 18.45 | 42.52 | 43.34 | 42.82 | 43.08 | 42.41 | 43.23 |
| | 6 | 19.28 | 19.21 | 19.25 | 19.38 | 19.37 | 19.23 | 42.76 | 43.30 | 43.17 | 43.14 | 42.88 | 43.42 |
| | 12 | 19.49 | 19.47 | 19.38 | 19.43 | 19.56 | 19.59 | 43.09 | 42.57 | 43.26 | 42.98 | 42.90 | 42.77 |
| | 24 | 19.76 | 19.85 | 19.73 | 19.81 | 19.76 | 19.84 | 43.02 | 42.92 | 43.04 | 42.84 | 42.94 | 42.88 |

Table 7: Perplexity and zero-shot accuracy of OPT-1.3B pruned to different sparsity types with Wanda and reconstructed in different settings with 1024 calibration samples on a single random seed. The best result for each setting is underlined, the best result for each sparsity type is highlighted in bold. SP, MP, DP: sparse, mixed, and dense propagation. CS: cosine similarity. ↓: lower is better, ↑: higher is better. Base perplexity of OPT-1.3B dense: 14.62, pruned unstructured: 18.31, pruned 2:4: 28.11.

| Sparsity type | Block size | Perplexity ↓ | | | | | | Zero-shot accuracy (in %) ↑ | | | | | |
| | | SP | | MP | | DP | | SP | | MP | | DP | |
| | | MSE | CS | MSE | CS | MSE | CS | MSE | CS | MSE | CS | MSE | CS |
| --- | --- | --- | --- | --- | --- | --- | --- | --- | --- | --- | --- | --- | --- |
| 50% unstructured | Per-matrix | 17.31 | 17.88 | 21.74 | 22.01 | 18.41 | 18.32 | 44.13 | 43.70 | 42.54 | 42.47 | 43.42 | 43.56 |
| | $\frac{1}{2}$ | 15.53 | 15.46 | 15.47 | **15.44** | 15.59 | 15.46 | 44.80 | **45.32** | 44.86 | 45.03 | 44.94 | 45.29 |
| | 1 | 16.02 | 15.91 | 16.42 | 16.54 | 16.09 | 16.13 | 44.16 | 44.80 | 44.05 | 44.30 | 44.27 | 44.91 |
| | 2 | 16.03 | 15.82 | 16.25 | 16.35 | 16.04 | 15.80 | 43.94 | 44.40 | 44.27 | 44.28 | 44.00 | 44.55 |
| | 6 | 16.12 | 16.17 | 16.17 | 16.25 | 16.14 | 16.20 | 44.47 | 44.36 | 44.30 | 44.19 | 44.45 | 44.34 |
| 2:4 | Per-matrix | 24.48 | 26.25 | 42.62 | 46.41 | 27.96 | 27.69 | 41.84 | 41.08 | 40.35 | 40.86 | 41.44 | 41.84 |
| | $\frac{1}{2}$ | 18.50 | 18.02 | **17.77** | 17.85 | 18.59 | 18.09 | 43.34 | 43.42 | 44.04 | **44.26** | 43.21 | 43.57 |
| | 1 | 19.72 | 18.42 | 19.40 | 19.78 | 19.86 | 18.92 | 42.73 | 43.52 | 43.54 | 43.18 | 42.76 | 43.63 |
| | 2 | 19.15 | 18.47 | 18.99 | 19.22 | 19.38 | 18.41 | 42.59 | 43.41 | 42.90 | 43.15 | 42.38 | 43.16 |
| | 6 | 19.33 | 19.24 | 19.48 | 19.71 | 19.41 | 19.26 | 42.75 | 43.09 | 42.63 | 42.66 | 42.89 | 43.41 |

Table 8: Perplexity and zero-shot accuracy of OPT-6.7B pruned to different sparsity types with Wanda and reconstructed in different settings with 1024 calibration samples on a single random seed. The best result for each setting is underlined, the best result for each sparsity type is highlighted in bold. SP, MP, DP: sparse, mixed, and dense propagation. CS: cosine similarity. ↓: lower is better, ↑: higher is better. Base perplexity of OPT-6.7B dense: 10.86, pruned unstructured: 12.06, pruned 2:4: 16.05.

| Sparsity type | Block size | Perplexity ↓ | | | | | | Zero-shot accuracy (in %) ↑ | | | | | |
| | | SP | | MP | | DP | | SP | | MP | | DP | |
| | | MSE | CS | MSE | CS | MSE | CS | MSE | CS | MSE | CS | MSE | CS |
| --- | --- | --- | --- | --- | --- | --- | --- | --- | --- | --- | --- | --- | --- |
| 50% unstructured | $\frac{1}{2}$ | 11.51 | **11.36** | 11.48 | 11.37 | 11.51 | **11.36** | 49.94 | 50.08 | 50.38 | 50.47 | 50.08 | 50.24 |
| | 1 | 11.64 | 11.63 | 11.92 | 11.88 | 11.58 | 11.69 | 49.92 | 50.62 | 50.03 | 50.20 | 49.82 | 50.52 |
| | 2 | 11.59 | 11.54 | 11.75 | 11.91 | 11.61 | 11.53 | 49.97 | 50.60 | 50.28 | 50.04 | 49.90 | 50.63 |
| | 8 | 11.85 | 11.89 | 11.89 | 11.95 | 11.84 | 11.90 | 49.84 | 49.76 | 50.03 | 49.90 | 49.76 | 49.78 |
| 2:4 | $\frac{1}{2}$ | 13.41 | 13.01 | 12.96 | **12.72** | 13.40 | 12.93 | 47.71 | 48.24 | 48.92 | **48.94** | 47.67 | 48.35 |
| | 1 | 13.75 | 13.08 | 13.77 | 13.56 | 13.55 | 13.22 | 47.70 | 48.77 | 48.31 | 48.75 | 47.82 | 48.76 |
| | 2 | 13.30 | 12.93 | 13.38 | 13.39 | 13.42 | 12.85 | 47.99 | 48.29 | 48.53 | 48.76 | 47.94 | 48.66 |
| | 8 | 13.49 | 13.45 | 13.56 | 13.57 | 13.54 | 13.45 | 47.93 | 48.14 | 48.07 | 48.28 | 47.86 | 48.09 |

Table 9: Perplexity and zero-shot accuracy of LLaMA-2-13B (40 transformer blocks) pruned to different sparsity types with Wanda and reconstructed in different settings with 1024 calibration samples on a single random seed. The best result for each setting is underlined, the best result for each sparsity type is highlighted in bold. SP, MP, DP: sparse, mixed, and dense propagation. CS: cosine similarity. ↓: lower is better, ↑: higher is better. Base perplexity of LLaMA-2-13B dense: 4.57, pruned unstructured: 5.54, pruned 2:4: 8.39.

| Sparsity type | Block size | Perplexity ↓ | | | | | | Zero-shot accuracy (in %) ↑ | | | | | |
| | | SP | | MP | | DP | | SP | | MP | | DP | |
| | | MSE | CS | MSE | CS | MSE | CS | MSE | CS | MSE | CS | MSE | CS |
| --- | --- | --- | --- | --- | --- | --- | --- | --- | --- | --- | --- | --- | --- |
| unstructured | $\frac{1}{2}$ | 5.33 | 5.33 | **5.26** | 5.33 | 5.34 | 5.35 | 60.53 | 60.84 | 60.99 | 60.96 | 60.59 | 60.59 |
| | 1 | 5.32 | 5.35 | **5.26** | 5.34 | 5.33 | 5.37 | 61.18 | 61.03 | 61.25 | **61.30** | 61.02 | 60.75 |
| | 2 | 5.32 | 5.36 | 5.27 | 5.33 | 5.33 | 5.37 | 60.93 | 60.45 | 60.91 | 60.84 | 60.62 | 60.61 |
| | 10 | 5.30 | 5.31 | 5.29 | 5.30 | 5.31 | 5.33 | 60.94 | 60.99 | 61.09 | 61.01 | 61.06 | 60.95 |
| 2:4 | $\frac{1}{2}$ | 6.76 | 6.80 | **6.24** | 6.59 | 6.71 | 6.80 | 56.16 | 56.15 | 58.66 | 58.30 | 56.34 | 56.10 |
| | 1 | 6.68 | 6.72 | 6.32 | 6.63 | 6.67 | 6.79 | 56.42 | 56.47 | **58.75** | 58.52 | 56.61 | 56.45 |
| | 2 | 6.56 | 6.64 | 6.35 | 6.52 | 6.60 | 6.73 | 55.94 | 56.35 | 57.81 | 58.38 | 56.93 | 56.33 |
| | 10 | 6.37 | 6.37 | 6.33 | 6.35 | 6.39 | 6.42 | 56.64 | 56.70 | 57.03 | 56.82 | 56.36 | 56.36 |

Table 10: Perplexity and zero-shot accuracy of LLaMA-3-8B (32 transformer blocks) pruned to different sparsity types with Wanda and reconstructed in different settings with 1024 calibration samples on a single random seed. The best result for each setting is underlined, the best result for each sparsity type is highlighted in bold. SP, MP, DP: sparse, mixed, and dense propagation. CS: cosine similarity. ↓: lower is better, ↑: higher is better. Base perplexity of LLaMA-3-8B dense: 5.83, pruned unstructured: 8.96, pruned 2:4: 21.58.

| Sparsity type | Block size | Perplexity ↓ | | | | | | Zero-shot accuracy (in %) ↑ | | | | | |
| | | SP | | MP | | DP | | SP | | MP | | DP | |
| | | MSE | CS | MSE | CS | MSE | CS | MSE | CS | MSE | CS | MSE | CS |
| unstructured | $\frac{1}{2}$ | 7.99 | 8.00 | 7.75 | 7.99 | 7.84 | 7.99 | 57.57 | 57.54 | **58.81** | 57.44 | 58.80 | 57.64 |
| | 1 | 7.76 | 8.00 | **7.70** | 7.92 | 7.75 | 8.02 | 58.55 | 57.77 | 58.08 | 58.45 | **58.84** | 57.66 |
| | 2 | 7.75 | 7.98 | 7.72 | 7.79 | 7.74 | 8.02 | 58.31 | 58.39 | 58.21 | 58.67 | 58.29 | 58.20 |
| | 8 | 7.82 | 7.89 | 7.86 | 7.83 | 7.83 | 7.95 | 58.07 | 57.94 | 57.86 | 58.04 | 58.10 | 57.76 |
| | 16 | 8.06 | 7.99 | 8.06 | 7.96 | 8.02 | 8.03 | 57.26 | 57.48 | 57.41 | 57.76 | 57.16 | 57.53 |
| 2:4 | $\frac{1}{2}$ | 15.79 | 15.28 | **9.96** | 11.96 | 11.36 | 12.78 | 46.91 | 47.58 | 55.74 | 50.71 | 51.18 | 49.66 |
| | 1 | 11.14 | 12.60 | 10.34 | 11.84 | 10.95 | 12.67 | 52.48 | 50.99 | 54.55 | **55.77** | 52.65 | 50.55 |
| | 2 | 10.74 | 11.90 | 10.34 | 11.18 | 10.56 | 12.04 | 52.00 | 51.24 | 52.78 | 53.33 | 51.51 | 50.16 |
| | 8 | 10.44 | 10.59 | 10.57 | 10.59 | 10.52 | 10.88 | 52.47 | 52.51 | 52.92 | 53.44 | 52.40 | 51.77 |
| | 16 | 11.10 | 10.90 | 11.18 | 10.92 | 11.11 | 11.02 | 51.81 | 52.07 | 51.83 | 52.55 | 51.16 | 51.60 |

Table 11: Perplexity and zero-shot accuracy of LLaMA-3-8B (32 transformer blocks) pruned to different sparsity types with Wanda and reconstructed in different settings with 1024 calibration samples from the MiniPile dataset on two random seed. The best result for each setting is underlined, the best result for each sparsity type is highlighted in bold. SP, MP, DP: sparse, mixed, and dense propagation. CS: cosine similarity. ↓: lower is better, ↑: higher is better. Base perplexity of LLaMA-3-8B dense: 5.83, pruned unstructured: 8.96, pruned 2:4: 21.58.

| Sparsity type | Block size | Perplexity ↓ | | | | | | Zero-shot accuracy (in %) ↑ | | | | | |
| | | SP | | MP | | DP | | SP | | MP | | DP | |
| | | MSE | CS | MSE | CS | MSE | CS | MSE | CS | MSE | CS | MSE | CS |
| unstructured | $\frac{1}{2}$ | 7.85 | 7.98 | 7.68 | 7.89 | 7.84 | 7.98 | 59.72 | 59.27 | 60.40 | **61.04** | 59.77 | 59.16 |
| | 1 | 7.73 | 7.92 | 7.68 | 7.83 | 7.72 | 7.94 | 59.31 | 59.55 | 60.04 | 60.23 | 59.90 | 59.40 |
| | 2 | 7.69 | 7.88 | **7.66** | 7.72 | 7.69 | 7.92 | 59.16 | 59.39 | 59.52 | 60.25 | 59.17 | 59.39 |
| | 8 | 7.71 | 7.77 | 7.75 | 7.72 | 7.72 | 7.83 | 58.81 | 58.87 | 59.00 | 59.15 | 58.68 | 58.59 |
| | 16 | 8.03 | 7.97 | 7.90 | 7.82 | 7.91 | 7.87 | 57.62 | 57.86 | 58.08 | 58.33 | 57.86 | 58.16 |
| 2:4 | $\frac{1}{2}$ | 11.48 | 12.77 | **9.65** | 10.75 | 11.16 | 12.33 | 53.18 | 51.16 | 56.50 | **56.66** | 53.23 | 51.05 |
| | 1 | 10.95 | 12.35 | 9.69 | 10.70 | 10.65 | 12.21 | 54.28 | 52.10 | 56.31 | 56.60 | 54.11 | 51.88 |
| | 2 | 10.60 | 11.58 | 10.26 | 10.83 | 10.44 | 11.57 | 52.73 | 51.69 | 54.05 | 55.18 | 53.06 | 51.02 |
| | 8 | 10.15 | 10.25 | 10.35 | 10.29 | 10.23 | 10.55 | 53.32 | 53.49 | 53.87 | 54.06 | 53.12 | 52.42 |
| | 16 | 11.82 | 10.46 | 10.82 | 10.52 | 10.77 | 10.61 | 50.24 | 52.00 | 52.80 | 53.48 | 52.27 | 52.29 |

Table 14: Zero-shot accuracies of OPT-6.7B pruned to different sparsity types with Wanda and reconstructed with MP and MSE on 1024 calibration samples on a single random seed.

| Sparsity type | Block size | Winogrande | RTE | OpenBookQA | HellaSwag | BoolQ | ARC-Easy | ARC-Challenge |
| --- | --- | --- | --- | --- | --- | --- | --- | --- |
| unstructured | $\frac{1}{2}$ | 64.48 | 53.79% | 26.80% | 48.59% | 67.37% | 65.07% | 29.95% |
| | 1 | 64.01% | 54.87% | 25.60% | 48.17% | 66.79% | 64.02% | 28.84% |
| | 2 | 64.80% | 53.79% | 27.00% | 48.18% | 67.52% | 64.27% | 29.01% |
| | 8 | 63.93% | 54.51% | 26.80% | 47.93% | 66.85% | 64.10% | 29.18% |
| 2:4 | $\frac{1}{2}$ | 62.27% | 55.60% | 26.20% | 46.71% | 65.38% | 62.50% | 27.30% |
| | 1 | 61.48% | 54.51% | 25.60% | 45.48% | 63.15% | 61.28% | 27.90% |
| | 2 | 62.67% | 54.87% | 26.40% | 45.50% | 64.01% | 61.15% | 26.96% |
| | 8 | 61.64% | 54.15% | 25.60% | 45.21% | 64.10% | 60.77% | 27.56% |

Table 12: Perplexity and zero-shot accuracy of four different models pruned to 50% unstructured and 2:4 sparsity with Wanda and reconstructed with MSE and MP. The best result for each (sparsity type, model) combination is highlighted in bold. ↓: lower is better, ↑: higher is better.

| Model Dense PPL | Block size | Perplexity ↓ | | Zero-shot accuracy (in %) ↑ | |
|---|---|---|---|---|---|
| | | unstructured | 2:4 | unstructured | 2:4 |
| | No rec. | 18.31 | 28.11 | 43.35 | 41.43 |
| OPT-1.3B | $\frac{1}{2}$ | **15.49** | **17.70** | **44.76** | **43.70** |
| 14.62 | 1 | 16.42 | 19.35 | 44.08 | 42.94 |
| | 2 | 16.26 | 19.02 | 44.26 | 42.82 |
| | 6 | 16.17 | 19.25 | 44.27 | 43.17 |
| | No rec. | 11.96 | 16.05 | 48.02 | 45.89 |
| OPT-6.7B | $\frac{1}{2}$ | **11.45** | **12.88** | **50.63** | **48.97** |
| 10.86 | 1 | 11.95 | 13.63 | 50.13 | 48.28 |
| | 2 | 11.82 | 13.36 | 50.32 | 48.51 |
| | 8 | 11.91 | 13.61 | 49.97 | 48.12 |
| | No rec. | 8.96 | 21.85 | 56.89 | 45.67 |
| LLaMA-3-8B | $\frac{1}{2}$ | 7.79 | **10.24** | **59.77** | **55.23** |
| 5.83 | 1 | **7.72** | 10.31 | 58.67 | 54.44 |
| | 2 | 7.75 | 10.32 | 58.36 | 52.72 |
| | 8 | 7.89 | 10.53 | 57.90 | 52.87 |
| | No rec. | 5.54 | 8.39 | 60.25 | 53.23 |
| LLaMA-2-13B | $\frac{1}{2}$ | **5.25** | **6.22** | 61.14 | **58.77** |
| 4.57 | 1 | **5.25** | 6.29 | **61.43** | 58.65 |
| | 2 | 5.27 | 6.35 | 60.86 | 57.84 |
| | 10 | 5.28 | 6.32 | 61.06 | 57.11 |

Table 13: Average zeroshot accuracy (in %) Qwen-2.5-32B when pruned with different pruning methods under four sparsity types and reconstructed with MP and MSE at block size $\frac{1}{2}$. The best result for each model and sparsity type is highlighted in bold.

| Model | Sparsity type | SparseGPT | | Wanda | | Magnitude | |
|---|---|---|---|---|---|---|---|
| | | No rec. | Rec. | No rec. | Rec. | No rec. | Rec. |
| Qwen-2.5-32B | 50% unstructured | 68.43 | **69.07** | 68.03 | 68.65 | 61.94 | 68.21 |
| | 60% unstructured | 66.12 | **67.26** | 65.78 | 66.67 | 55.24 | 66.66 |
| | 2:4 | 66.33 | 66.79 | 66.41 | **67.59** | 58.55 | 67.06 |
| | 4:8 | 67.44 | 67.83 | 67.62 | **68.56** | 59.49 | 68.20 |

Table 15: Zero-shot accuracies of LLaMA-2-13B pruned to different sparsity types with Wanda and reconstructed with MP and MSE on 1024 calibration samples on a single random seed.

| Sparsity type | Block size | Winogrande | RTE | OpenBookQA | HellaSwag | BoolQ | ARC-Easy | ARC-Challenge |
|---|---|---|---|---|---|---|---|---|
| unstructured | $\frac{1}{2}$ | 71.74 | 68.23% | 33.60% | 57.62% | 80.83% | 76.39% | 43.26% |
| | 1 | 71.59% | 68.59% | 33.80% | 57.55% | 80.73% | 76.39% | 42.83% |
| | 2 | 71.27% | 67.87% | 33.20% | 57.62% | 80.12% | 76.64% | 42.66% |
| | 10 | 70.96% | 68.23% | 33.60% | 57.29% | 80.73% | 77.06% | 42.58% |
| 2:4 | $\frac{1}{2}$ | 69.46% | 64.62% | 31.60% | 54.75% | 78.90% | 73.02% | 38.74% |
| | 1 | 68.98% | 65.70% | 30.60% | 54.57% | 79.42% | 73.61% | 38.91% |
| | 2 | 69.69% | 66.43% | 30.60% | 53.93% | 78.59% | 73.65% | 38.14% |
| | 10 | 67.88% | 62.45% | 30.00% | 53.49% | 77.00% | 73.61% | 38.40% |

Table 16: Zero-shot accuracies of LLaMA-3-8B pruned to different sparsity types with Wanda and reconstructed with MP and MSE on 1024 calibration samples on a single random seed.

| Sparsity type | Block size | Winogrande | RTE | OpenBookQA | HellaSwag | BoolQ | ARC-Easy | ARC-Challenge |
|---|---|---|---|---|---|---|---|---|
| unstructured | $\frac{1}{2}$ | 69.93% | 55.96% | 29.20% | 55.57% | 77.46% | 76.64% | 44.03% |
| | 1 | 70.17% | 55.60% | 29.00% | 55.03% | 74.89% | 76.85% | 44.71% |
| | 2 | 68.98% | 62.82% | 28.40% | 54.07% | 78.84% | 75.72% | 43.17% |
| | 8 | 69.38% | 63.54% | 29.40% | 54.12% | 78.69% | 75.97% | 43.43% |
| | 16 | 69.61% | 61.01% | 29.00% | 54.42% | 75.47% | 76.05% | 43.17% |
| 2:4 | $\frac{1}{2}$ | 64.80% | 58.12% | 26.60% | 49.99% | 69.66% | 68.69% | 35.58% |
| | 1 | 64.25% | 62.82% | 26.20% | 49.07% | 68.69% | 68.94% | 35.67% |
| | 2 | 65.98% | 55.60% | 25.00% | 47.40% | 67.06% | 65.57% | 33.28% |
| | 8 | 65.67% | 60.65% | 25.40% | 47.43% | 70.34% | 66.37% | 34.13% |
| | 16 | 63.22% | 57.40% | 24.60% | 47.97% | 69.45% | 67.76% | 34.13% |

Table 17: Perplexity and zero-shot accuracy statistics for each block size for OPT-1.3B and 10 random seeds.

| Block size | Perplexity (PPL) | STD PPL | Max - min PPL | Zero-shot accuracy (ZS) | STD ZS | Max - min ZS |
|---|---|---|---|---|---|---|
| $\frac{1}{2}$ | 15.37 | 0.08 | 0.30 | 44.26% | 0.0013 | 0.43% |
| 1 | 16.29 | 0.14 | 0.52 | 44.07% | 0.0014 | 0.44% |
| 2 | 16.15 | 0.15 | 0.47 | 43.62% | 0.0029 | 0.99% |
| 6 | 16.11 | 0.08 | 0.26 | 43.81% | 0.0011 | 0.32% |
| 12 | 16.18 | 0.06 | 0.19 | 44.12% | 0.0009 | 0.29% |
| 24 | 16.24 | 0.07 | 0.20 | 43.82% | 0.0019 | 0.59% |

Figure 5: Deviation of perplexity of OPT-1.3B from the mean perplexity of 10 random seeds for each block size. The deviation is normalized to the range from minimum to maximum perplexity for each block size.

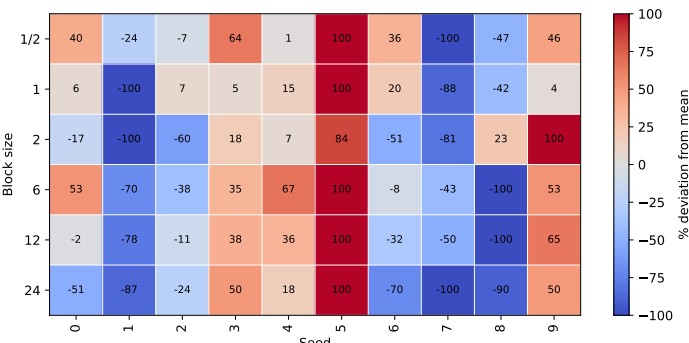

## C  HYPERPARAMETERS

Table 18: Hyperparameters used for local reconstruction, full retraining, and Mask-LoRA fine-tuning.

| Method | Learning rate | Warmup ratio | Scheduler | Number of epochs | Optimizer |
|---|---|---|---|---|---|
| Local reconstruction | [1e-6, 1e-3] | 0.1 | Linear | $\{1, 5, 10, 20\}$ | AdamW |
| Mask-LoRA fine-tuning | [1e-7, 1e-3] | 0.1 | Linear | $\{1, 5, 10\}$ | AdamW |
| Full retraining | [1e-7, 1e-3] | 0.1 | Linear | $\{1, 5, 10, 20\}$ | AdamW |

| Method | Batch size | Gradient accumulation | LoRA rank | LoRA alpha | Max. gradient norm |
|---|---|---|---|---|---|
| Local reconstruction | 2 | 1 | - | - | - |
| Mask-LoRA fine-tuning | 1 | 2 | 16 | 32 | 1 |
| Full retraining | 1 | 2 | - | - | 1 |