# OpenReview forum: "A Free Lunch in LLM Compression: Revisiting Retraining after Pruning"
_ICLR.cc/2026/Conference — Submitted to ICLR 2026_

### Official Review · Reviewer_in2y · 2025-10-27

**Soundness:** 2
**Presentation:** 2
**Contribution:** 2
**Rating:** 2
**Confidence:** 4

**Summary:**

The paper studies the performance of retraining after pruning under different setups and tries to explore the optimal setup for achieving the best performance. The paper provides experimental evidence that simple pruning methods like Wanda can achieve even better performance than other more complex pruning methods when combined with a good retraining strategy.

**Strengths:**

The retraining problem after pruning is worth studying. The paper considers different cases for retraining across various inputs, targets, and pipeline setups, and shows experimental results comparing the performance under different setups.

**Weaknesses:**

- My main concerns lie in the presentation of this paper. I don’t mean that the paper contains a lot of mistakes, but the overall presentation—such as the definition of equations and terms—makes it quite hard to understand. See the Questions section for some examples.
- I’m also concerned about the contribution of this paper. Most of the key statements are already well-known from previous work. For example, the paper shows that per-matrix reconstruction consistently underperforms. However, this is already a well-known result and, in fact, per-matrix reconstruction is not a commonly adopted formulation in practice.
- I’m wondering about the rationale for combining losses at different levels. Would performance improve if we selectively used certain losses instead? For example, the paper combines per-matrix reconstruction loss, attention component loss, and MLP loss—similar to what was explored in [1] by NVIDIA, which also investigated retraining strategies for pruned models and gave much convincing conclusions.
- The experiments mainly focus on perplexity. However, perplexity alone cannot fully represent a model’s performance after retraining. It’s expected that retraining on next-word prediction will lead to improvements on that same task. What really matters is the model’s generalization ability on downstream tasks. Unfortunately, I did not see sufficient experimental coverage in this regard.

Overall, I think the presentation, contributions, and experimental setup need to be improved before the paper is ready for acceptance.

[1] Compact Language Models via Pruning and Knowledge Distillation

**Questions:**

- Line 189: It’s hard for me to capture the idea quickly. Some matrices are bolded and others are not. Also, in Line 195, should the X actually be \hat{X}?
- Line 220: Terms like “block size one-half” are hard for the reader to understand at a glance.
- Later in the paper, there are many terms like block size 1/2, 1, 12, which are also difficult to parse quickly. It’s unclear to the reader what exactly these refer to, making it harder to follow the main ideas.
- The differences in the results are quite limited and not sufficient to support strong conclusions. For example, in Table 3, the perplexity differences between Wanda and SparseGPT are mostly within 0.1. It’s difficult to justify the claim of “outperformance” with such a small margin. If you change the seed for calibration data selection or adjust the batch size, the conclusion could easily be reversed.

---

> ### Author Response · Authors · 2025-11-21
>
> We thank the reviewer for the detailed and constructive comments on clarity, presentation, and evaluation coverage. We address the specific points below.
>
> Regarding presentation clarity, we appreciate the feedback and will revise Section 2 to improve notation consistency and readability. This includes correcting the bold formatting issue, providing clearer definitions of reconstruction granularities, and using consistent terminology rather than shorthand such as block size 1/2.
>
> The reviewer notes that “per-matrix reconstruction consistently underperforms” is a known fact. Our contribution lies in performing, to the best of our knowledge, the first controlled study that isolates the reconstruction granularity as an independent variable under identical pruning, data set, and optimization settings, allowing for clear comparison across granularities and models. Prior work has noted the weaknesses of per-matrix reconstruction due to error propagation, but has not experimentally evaluated granularity in a fully controlled setting in the context of LLM reconstruction. The finding that all granularities (ranging from separate reconstruction of attention and MLP components to joint reconstruction of multiple transformer blocks) behave nearly identically in final performance, despite very large differences in memory requirements, and that fine-grained reconstruction allows operation on hardware where coarse-grained reconstruction is infeasible, is novel and practically relevant. For example, take LLaMA-2-13B, which, after pruning, cannot be fine-tuned with LoRA (rank 16) on a single A100 80GB, but local reconstruction of one transformer block at a time is possible on a single GPU. This challenges the widely assumed requirement that retraining should be avoided in the LLM pruning setting and provides actionable guidance for researchers and practitioners working in resource-constrained environments.
>
> Regarding the point that “the performance differences in Table 3 are mostly within 0.1 perplexity”, while these differences are small in absolute magnitude, they are consistent between models, sparsity levels, and multiple seeds, and they invert the usual ordering between pruning methods: before reconstruction SparseGPT outperforms Wanda, but after reconstruction Wanda + reconstruction consistently matches (eve outperforms) SparseGPT + reconstruction, indicating that the additional algorithmic complexity of second-order updates in SparseGPT does not translate into advantages under local reconstruction.
>
> Regarding “combining losses at different levels”, we agree that this is an interesting question. In contrast to the referenced NVIDIA distillation work, which operates in a full-model distillation context, local reconstruction targets scenarios where full teacher access or distillation is infeasible. Although exploration of intermediate-activation constraints is promising, it is outside the scope of the current controlled granularity study and represents future work.
>
> On evaluation breadth, we will include the per-task zero-shot accuracy results underlying our aggregated zero-shot accuracy metric in the appendix. These results demonstrate that the trends observed in perplexity and average zero-shot accuracy align with downstream generalization performance.

---

> > ### Comment · Reviewer_in2y · 2025-11-26
> >
> > Thank you for the authors’ response. I would like to maintain my current score at this point, but I would be happy if the authors could include further results and improve the presentation.

---

> > > ### Author Response · Authors · 2025-12-03
> > >
> > > We have implemented some of the feedback to improve notation consistency and readability in Section 2 in the revised version.
> > >
> > > The appendix now contains tables (15 and 16) detailing the zero-shot accuracies on the tasks Winogrande, RTE, OpenBookQA, HellaSwag, BoolQ, ARC-Easy, and ARC-Challenge for local reconstruction under different sparsity types and at different granularities.

---

### Official Review · Reviewer_nWCe · 2025-10-27

**Soundness:** 3
**Presentation:** 3
**Contribution:** 3
**Rating:** 6
**Confidence:** 4

**Summary:**

This paper re-examines the post-pruning reconstruction process in large language models by systematically investigating design choices including propagation strategies, loss functions, and reconstruction granularities. The study reveals several findings that challenge conventional wisdom, particularly that reconstructing attention and MLP components separately within each transformer block (block size 1/2) achieves a favorable balance between resource efficiency and model performance. This approach not only requires less memory but also demonstrates competitive or superior performance in perplexity and zero-shot accuracy compared to full model retraining in certain scenarios.

**Strengths:**

1.Comprehensive research design covering comparisons of multiple propagation strategies, loss functions, and reconstruction granularities, providing systematic experimental analysis.

2.Extensive experimental scope across multiple mainstream models (OPT-1.3B, OPT-6.7B, LLaMA-2-13B, LLaMA-3-8B), enhancing the generalizability of findings.

3.Identification of block size 1/2 reconstruction as a favorable balance point between resource efficiency and performance, offering practical guidance for applications.

**Weaknesses:**

1.When the reconstruction granularity is small, more iterative training is required, so comparing only peak memory usage is unfair; training time comparisons should also be included.

2.This paper demonstrates through detailed experiments that a block size of 1/2 yields the best results, but it lacks certain theoretical explanations.

**Questions:**

1.Could the authors provide some insights to explain why a block size of 1/2 achieves optimal performance? This would strengthen the paper's contributions and provide deeper understanding.

---

> ### Author Response · Authors · 2025-11-21
>
> We thank the reviewer for constructive feedback and for recognizing both the systematic scope and practical relevance of our study.
>
> Regarding training-time comparisons, we agree that reporting wall-clock time is useful for completeness, and we will include these measurements in the revised version. However, we emphasize that local reconstruction is intended for settings where full retraining or LoRA is not feasible due to memory constraints. For example, for LLaMA-2-13B, full retraining after pruning is not possible on a single A100 80G GPU, but reconstruction of one transformer block at a time remains possible. As with many memory-efficient training approaches, reduced memory usage inherently trades off against increased compute time, and the primary value of local reconstruction lies in enabling post-pruning recovery for large models on constrained hardware rather than minimizing wall-clock time. The added results will clarify this trade-off transparently.
>
> Regarding mathematical intuition, let $W\in\mathbb R^{d_{out}\times d_{in}}$ be a weight matrix, $M\in\{0,1\}^{d_{out}\times d_{in}}$ a pruning mask, and $(x_1,...,x_B)=X\in\mathbb R^{d_{in}\times B}$ a calibration set. Then, the per-matrix reconstruction problem is $$\min_{\tilde W}\|(W-M\odot \tilde W)X\|_F^2.$$ Since $\|(W-M\odot \tilde W)X\|_F^2=\text{trace}((W-M\odot \tilde W)XX^\top(W-M\odot \tilde W)^\top)$, the objective only depends on the Gram matrix $G=XX^\top$ which stabilizes rather quickly when the number of datapoints is increased. This means that, no matter how large the dataset is, the problem effectively sees only $d$ orthogonal directions (the eigenvectors of $G$) weighted by their variances. Additionally, when the pruned layer does not have enough degrees of freedom (which is generally the case), improving the fit in one direction necessarily worsens it in another, i.e., there is no place to "hide errors". In contrast, when reconstructing a nonlinear submodel such as the MLP in a transformer block, the capactity is high enough to "hide errors". Empirically, we see that at block size 1/2, we reach the capacity needed to fit the calibration data well, and with 1k samples, increasing the capacity (the size of the reconstructed submodels) does not give us any advantage.

---

> ### Comment · Reviewer_nWCe · 2025-11-23
>
> Thank you for the author's detailed response. I maintain my score to support the acceptance of this paper.

---

### Official Review · Reviewer_5pCu · 2025-10-27

**Soundness:** 2
**Presentation:** 2
**Contribution:** 2
**Rating:** 2
**Confidence:** 4

**Summary:**

The paper conducts an extensive study across multiple LLMs, exploring how different reconstruction settings affect post-pruning recovery. By isolating factors such as propagation strategy, loss function, and reconstruction granularity, the work provides valuable empirical insights into how local reconstruction actually works in practice. Demonstrating that simple pruning methods like Wanda can outperform more complex schemes when coupled with proper reconstruction offers a practical contribution.

**Strengths:**

A thorough empirical study investigates the impact of various reconstruction configurations on post-pruning performance recovery. I appreciate the paper’s well-defined experiments and its simple yet effective findings. My main concern lies in the level of novelty, though I believe this point would be best discussed among reviewers and potential readers.

**Weaknesses:**

- Although the paper provides strong empirical evidence, its experimental novelty is somewhat limited. The work mainly explores reconstruction granularity variations rather than introducing a new methodological direction.
- The pruning methods considered feel narrow in scope (only unstructured and semi-structured pruning). Including structured pruning experiments would strengthen the paper’s generality and practical relevance.
- The statement "Reconstructing at block size 1/2 is most effective." (page 6) appears slightly overstated. Although this trend holds for OPT models, the LLaMA series exhibits comparable or even different optimal settings, as already noted in the paper.
- The calibration data appear to be limited to sequences sampled from C4. It would be valuable to analyze how varying the type of calibration data (e.g., using a different dataset or an instruction-tuning corpus) affects reconstruction performance.
- When we refer to retraining after pruning for performance recovery, we usually mean retraining the model on a next-token prediction task using logits with a sufficient amount of data. However, I think the term full retraining in this paper does not correspond to that setting. Moreover, it is difficult to find a clear and direct comparison between the proposed reconstruction scheme and the kind of fully retrained model that we usually mean by retraining.
- It would be valuable to include a comparison or discussion of other regression-based performance recovery or compensation methods in LLM pruning.
  * SlimLLM: Accurate Structured Pruning for Large Language Models ( https://arxiv.org/abs/2505.22689 )
  * Olica: Efficient Structured Pruning of Large Language Models without Retraining ( https://arxiv.org/abs/2506.08436 )
  * Fluctuation-based Adaptive Structured Pruning for Large Language Models ( https://arxiv.org/abs/2312.11983 )

**Questions:**

Please refer to the Weakness part.

---

> ### Author Response · Authors · 2025-11-21
>
> We thank the reviewer for the thoughtful evaluation and for highlighting both strengths and areas for improvement. We appreciate the positive assessment of the empirical depth and practical contribution of our study. In the following, we address the concerns raised.
>
> Regarding novelty, the key contribution of this work is twofold. First, we provide a controlled and systematic study that isolates local reconstruction granularity (how coarse or fine we divide the model for separate reconstruction of submodels) as an independent factor across architectures, pruning methods, and evaluation settings. Second, our results reveal a previously unreported and practically meaningful finding: across all evaluated models, all granularities at or above block size 1/2 (separate reconstruction of attention and MLP components) achieve comparable post-pruning performance, while finer granularities require dramatically less memory, enabling reconstruction of large models on limited hardware without sacrificing accuracy. For example, take LLaMA-2-13B, which, after pruning, cannot be fine-tuned with LoRA (rank 16) on a single A100 80GB, but local reconstruction of one transformer block at a time is possible on a single GPU. This challenges the widely assumed requirement that retraining is to be avoided in the LLM pruning setting and provides actionable guidance for researchers and practitioners working in resource-constrained environments.
>
> Regarding the reviewers concerns about "the pruning methods considered feel narrow in scope", we agree that including structured settings strengthens practical relevance. Additional experiments comparing structured Wanda + local reconstruction against FLAP structured pruning are underway, including downstream accuracy evaluation, and will be included in the revised paper.
>
> On the statement “block size 1/2 is most effective”, we will reword to improve clarity. Our intent is to report that all granularities (except reconstructing each matrix separately) perform similarly in accuracy, but block size 1/2 is the finest granularity that achieves this performance level, making it the most resource-efficient option under memory constraints without performance trade-off.
>
> Regarding data set diversity, we agree that this is important. We are running experiments using the Pile dataset, and results will be included in the appendix of the revision. These will help to examine the sensitivity to calibration distribution, as suggested.
>
> On full retraining baselines, we acknowledge the mismatch between the term “full retraining” in our current wording and its typical meaning in the literature. Additional full retraining experiments using substantially larger datasets are underway. Preliminary results show that full retraining takes 128x more samples to reach perplexity comparable to fine-grained local reconstruction on 1k samples.

---

> ### Comment · Reviewer_5pCu · 2025-11-27
>
> I appreciate the authors’ response. While some clarifications were helpful, several experiments addressing my and other reviewers’ comments are still incomplete. Therefore, I will maintain my score at this moment.

---

> ### Author Response · Authors · 2025-11-27
>
> Thank you for your response. We uploaded a revised version of the paper.
>
> We have added experiments with Mask-LoRA to the paper, showing that for LLaMA-3-8B. The table shows that, with 128 times more samples and 27.2 times more compute time, Mask-LoRA [1] can match the performance of local reconstruction.
>
> |         Method        | Number of samples | Perplexity |  Compute Time |
> |:---------------------:|:-----------------:|:----------:|:-------------:|
> |  Local reconstruction |        1024       |    7.72    | 42.36 minutes |
> | Mask-LoRA fine-tuning |       131072      |    7.70    |  19.17 hours  |
>
> Furthermore, we extended the full retraining of OPT-1.3B (Figure 1) to 131k samples, showing that full retraining fails to surpass fine-grained local reconstruction even with 128x more samples.
>
> The appendix now contains a table showing results for the reconstruction of LLaMA-3-8B on the MiniPile data set. The results are almost identical to the ones obtained from experiments on the C4 data set.
>
> We added a table comparing structured Wanda with and without local reconstruction to FLAP on LLaMA-2-7B. Interestingly, structured Wanda is able to outperform FLAP after reconstruction, even though FLAP non-uniformly distributes the sparsity across layers, making it a much more complex approach.
> |          | Perplexity |    Perplexity    |   Perplexity  | Zero-shot accuracy | Zero-shot accuracy | Zero-shot accuracy |
> |:--------:|:----------:|:----------------:|:-------------:|:------------------:|:------------------:|:------------------:|
> | Sparsity |    FLAP    | Wanda-sp no rec. | Wanda-sp rec. |        FLAP        |  Wanda-sp no rec.  |    Wanda-sp rec    |
> |   20\%   |    7.81    |       8.61       |      7.76     |       53.42\%      |       54.40\%      |       56.09\%      |
> |   50\%   |    46.25   |       57.77      |     30.97     |       42.87\%      |       36.05\%      |       44.24\%      |
>
>
> [1] Zimmer, Max, et al. "Perp: Rethinking the prune-retrain paradigm in the era of LLMs." arXiv preprint arXiv:2312.15230 (2023).

---

### Official Review · Reviewer_mQkh · 2025-10-28

**Soundness:** 2
**Presentation:** 2
**Contribution:** 1
**Rating:** 2
**Confidence:** 4

**Summary:**

The paper studies “partial training” for LLMs where only selected components are optimized while the rest are reconstructed or kept fixed. It claims that partial training outperforms full-parameter training with lower memory and compute, and shows results on small to mid-size models and standard language modeling metrics.

**Strengths:**

- Reports an observation that partial training can beat full-parameter training on some setups. This is interesting and, if robust, practically useful.
- Simple pipeline. Minimal code changes in principle.

**Weaknesses:**

- **Model scale and diversity.** Experiments are limited to small or mid-size models. No validation on modern ≥13B–70B models. Generality is unproven.
- **Missing LoRA and strong PEFT baselines.** No comparison to LoRA/QLoRA, adapters, prefix/prompt tuning, or modern pruning/distillation. The central claim “better than full training” must also be “non-trivially competitive with PEFT,” which is not shown.
- **Hyperparameter opacity.** Training schedules, LR ranges, warmup, weight decay, gradient clipping, batch sizes, number of steps/epochs, and early-stopping criteria are under-specified. Reproducibility is weak.
- **Dataset coverage.** Only a narrow calibration/training distribution is used. No dataset ablation. Distribution shift is not analyzed.
- **Training dynamics absent.** No loss/valid curves, gradient norms, or representation-shift diagnostics to justify the claim.
- **Compute and memory accounting.** Comparisons focus on peak memory during the proposed procedure, not end-to-end FLOPs, wall-clock, or activation I/O when reconstructing with the dense teacher. This weakens “cheaper than full training.”
- **Seeds and variance.** Several tables, i.e., Table 1-3, appear single-seed. No mean±std. Claim strength is overstated.
- **Over-claiming.** Abstract and conclusions generalize beyond the tested setting. Baseline tuning appears narrow and may be under-optimized, yet conclusions are broad.

**Questions:**

1. More experiments as noted above: larger models, more datasets, PEFT baselines, multi-seed with mean±std, full compute/memory accounting.
2. Please report training dynamics: train/valid loss over steps, PPL vs steps, gradient norms, layerwise updates, and parameter movement norms.
3. Provide mathematical intuition: what objective is the partial training implicitly optimizing relative to full training? Is there a projection, subspace, or spectral alignment view that explains when partial training wins?
4. How are components selected for optimization vs reconstruction? Any criterion beyond engineering convenience (e.g., sensitivity, curvature, Fisher, activation energy)?
5. How many steps and samples until partial training surpasses full training? Is the effect transient or stable after long training?

---

> ### Author Response · Authors · 2025-11-21
>
> We thank the reviewer for the careful evaluation and constructive comments that have helped us improve the clarity and positioning of our work. We address the main points below.
>
> Regarding experimental scale and baselines, we have begun experiments on 32B-scale models, LoRA fine-tuning after pruning, and the Pile dataset, and will include the results in the revised version. Preliminary observations indicate that LoRA requires at least 128x more samples than local reconstruction (with 1k calibration samples) to reach comparable post-pruning performance. We emphasize that this result is distinct from our general motivation: local reconstruction remains feasible for large models on limited hardware, as only the currently reconstructed block needs to fit in memory. For example, while a LLaMA-2-13B model cannot be fine-tuned with LoRA (rank 16) on a single A100 80GB GPU, while block size 1 reconstruction (separate reconstruction of each transformer block) still fits and runs without model parallelism.
>
> Regarding hyperparameters and reproducibility, learning rate ranges, schedule, and warm-up, epochs, calibration set size, and batch size are stated in the first paragraph section 3. For reconstruction, we do not clip gradients or stop early. We will add a hyperparameter table detailing our setups for local reconstruction, full retraining, and LoRA fine-tuning to the appendix. Tables 1-3 and Figures 1 and 3 report averages across multiple seeds, as stated in the first paragraph of section 3. A random seed sensitivity analysis will be added to the revised paper.
>
> Regarding compute, memory, and training dynamics, we will include end-to-end training-time comparisons between full fine-tuning, LoRA, and local reconstruction, as well as train/validation loss curves. These results will illustrate the trade-off between memory and compute: while local reconstruction substantially reduces memory requirements, this comes at the cost of additional training time compared to full-model training on the same calibration set.
>
> We will revise the abstract and conclusion to clarify that jointly reconstructing multiple transformer blocks exhibits a similar final model performance as reconstructing attention and MLP separately (block size 1/2), but block size 1/2 requires far less memory. This makes it the most practical choice in memory-constrained settings, as it minimizes VRAM usage while matching or exceeding coarser approaches. Experiments on Qwen-2.5-32B-Instruct will support the generality of our claim.
>
> Regarding component selection, we clarify that, in practice, granularity should be chosen based on hardware constraints: the size of the submodel should be chosen such that local reconstruction is feasible on given hardware. Take the example from above; a 13B parameter LLaMA model can not be reconstructed at block size 10 with a single A100 80GB GPU, but at block size 2 this it can.
>
> Lastly, regarding mathematical intuition, let $W\in\mathbb R^{d_{out}\times d_{in}}$ be a weight matrix, $M\in\{0,1\}^{d_{out}\times d_{in}}$ a pruning mask, and $(x_1,...,x_B)=X\in\mathbb R^{d_{in}\times B}$ a calibration set. Then, the per-matrix reconstruction problem is $$\min_{\tilde W}\|(W-M\odot \tilde W)X\|_F^2.$$ Since $\|(W-M\odot \tilde W)X\|_F^2=\text{trace}((W-M\odot \tilde W)XX^\top(W-M\odot \tilde W)^\top)$, the objective only depends on the Gram matrix $G=XX^\top$ which stabilizes rather quickly when the number of datapoints is increased. This means that, no matter how large the dataset is, the problem effectively sees only $d$ orthogonal directions (the eigenvectors of $G$) weighted by their variances. Additionally, when the pruned layer does not have enough degrees of freedom (which is generally the case), improving the fit in one direction necessarily worsens it in another, i.e., there is no place to "hide errors". In contrast, when reconstructing a nonlinear submodel such as the MLP in a transformer block, the capactity is high enough to "hide errors". Empirically, we see that at block size 1/2, we reach the capacity needed to fit the calibration data well, and with 1k samples, increasing the capacity (the size of the reconstructed submodels) does not give us any advantage.

---

> > ### Author Response · Authors · 2025-12-03
> >
> > We added experiments with Qwen-2.5-32B to tables 2 and 5, comparing both different granularities and pruning methods. As shown in the table below, the difference in performance between the different granularities stays negligible even for Qwen-2.5-32B.
> > | Block size |  Perplexity  | Perplexity | Zero-shot accuracy | Zero-shot accuracy |
> > |:----------:|:------------:|:----------:|:------------------:|:------------------:|
> > |            | unstructured |     2:4    |    unstructured    |         2:4        |
> > |   No rec.  |     6.09     |    8.00    |        68.03       |        66.41       |
> > |     1/2    |     5.93     |    6.06    |        69.05       |        68.79       |
> > |      1     |     5.92     |    5.97    |        69.18       |        69.06       |
> > |      2     |     5.98     |    6.01    |        69.15       |        69.19       |
> > |      8     |     5.97     |    7.07    |        69.09       |        68.86       |
> >
> > We have added a table comparing local reconstruction and Mask-LoRA [1]. The table shows that, with 128 times more samples and 27.2 times more compute time, Mask-LoRA can match the performance of local reconstruction.
> > |         Method        | Number of samples | Perplexity |  Compute Time |
> > |:---------------------:|:-----------------:|:----------:|:-------------:|
> > |  Local reconstruction |        1024       |    7.72    | 42.36 minutes |
> > | Mask-LoRA fine-tuning |       131072      |    7.70    |  19.17 hours  |
> >
> > Regarding the reviewers' concerns about dataset coverage, Table 11 in the appendix now shows results for reconstruction at different granularities, propagation strategies, and loss functions for LLaMA-3-8B on the MiniPile data set. The results are almost identical to the ones obtained from experiments on the C4 data set.
> >
> > We have added a table detailing hyperparameter configurations to the appendix and referenced it in the experimental setup section. Our hyperparameter configurations are as follows (for the comparison of local reconstruction and Mask-LoRA, both were run with a maximum of 10 epochs).
> > |         Method        | Learning rate |      Warmup ratio     | Scheduler | Number of epochs |      Optimizer     |
> > |:---------------------:|:-------------:|:---------------------:|:---------:|:----------------:|:------------------:|
> > |  Local reconstruction |  [1e-6, 1e-3] |          0.1          |   Linear  |    {1,5,10,20}   |        AdamW       |
> > | Mask-LoRA fine-tuning |  [1e-7, 1e-3] |          0.1          |   Linear  |     {1,5,10}     |        AdamW       |
> > |    Full retraining    |  [1e-7, 1e-3] |          0.1          |   Linear  |    {1,5,10,20}   |        AdamW       |
> >
> > |         Method        |   Batch size  | Gradient accumulation | LoRA rank |    LoRA alpha    | Max. gradient norm |
> > |:---------------------:|:-------------:|:---------------------:|:---------:|:----------------:|:------------------:|
> > |  Local reconstruction |       2       |           1           |     -     |         -        |          -         |
> > | Mask-LoRA fine-tuning |       1       |           2           |     16    |        32        |          1         |
> > |    Full retraining    |       1       |           2           |     -     |         -        |          1         |
> >
> > Regarding the reviewer's last question, "How many steps and samples until partial training surpasses full training? Is the effect transient or stable after long training?", we have extended Figure 3 (qualitative comparison of reconstruction approaches on OPT-1.3B) to show full retraining up to 131k samples. It is still far from surpassing block size 1/2 (attention and MLP separately) reconstruction at 8192 samples (the maximum number of samples we experimented with for local reconstruction). Furthermore, the second table above shows that Mask-LoRA fine-tuning with 131k samples only matches local reconstruction with 1024 samples.
> >
> > [1] Zimmer, Max, et al. "Perp: Rethinking the prune-retrain paradigm in the era of LLMs." arXiv preprint arXiv:2312.15230 (2023).

---

### Meta-Review · Area_Chair_Wioe · 2026-01-07

**Summary:**

This paper investigates the key design choices involved in reconstructing or retraining the remaining weights after pruning. The main observation is that reconstructing the attention and MLP components separately within each transformer achieves comparable model performance to more coarse-grained reconstructions, while also being the most resource-efficient. However, significant concerns remain regarding insufficient empirical verification, overstated claims, and novelty, which collectively weaken the contribution of this work. As a result, the paper is not recommended for acceptance.

**Reviewer Concerns:**

**Addressed Concerns:**

1.	**Details on the experimental setting and some writing issues (Reviewer mQkh and 5pCu).** The authors provided some additional details on experimental setting, such as values of hyperparameters. They also clarify some writing issues such as the  mismatch between the term “full retraining” in this paper and its typical meaning in the literature.

**Outstanding Concerns:**

1.	**Novelty and overstated claims (Reviewer mQkh and in2y).** Most of the key statements are already well-known from previous work. For example, the paper shows that per-matrix reconstruction consistently underperforms.

2.	**Insufficient experimental verification (Reviewer mQkh, 5pCu and in2y).** Although the authors have provided some additional experimental results in the rebuttal, they are still incomplete according to the comments from the reviewers.

Moreover, the empirical conclusions presented by the authors may heavily depend on the experimental setup, including factors such as dataset size and hyperparameter configurations. Consequently, in the absence of sufficient theoretical justification, these conclusions may lack reliability.

**Reviewer Scores:**

- **Reviewer  mQkh:** The score is likely to remain unchanged (2) due to the overstated claims and insufficient empirical verification.

- **Reviewer  5pCu:** The score is likely to remain unchanged (2) as the authors only conduct experiments with unstructured and semi-structured pruning methods.

- **Reviewer  nWCe:** The score remains unchanged (6) as indicated by the reviewer.

- **Reviewer  in2y:** The score remains unchanged (2) due to the writing issues and already well-known statements.

---

### Decision · Program_Chairs · 2026-01-26

Reject